# GRAPH KERNEL NEURAL NETWORKS

## ABSTRACT

The convolution operator at the core of many modern neural architectures can effectively be seen as performing a dot product between an input matrix and a filter. While this is readily applicable to data such as images, which can be represented as regular grids in the Euclidean space, extending the convolution operator to work on graphs proves more challenging, due to their irregular structure. In this paper we propose to use graph kernels, i.e., kernel functions that compute an inner product on graphs, to extend the standard convolution operator to the graph domain. This allows us to define an entirely structural model that does not require computing the embedding of the input graph. Our architecture allows to plug-in any type and number of graph kernels and has the added benefit of providing some interpretability in terms of the structural masks that are learned during the training process, similarly to what happens for convolutional masks in traditional convolutional neural networks. We perform an extensive ablation study to investigate the impact of the model hyper-parameters and we show that our model achieves competitive performance on standard graph classification datasets.

## 1 INTRODUCTION

In recent years, graph neural networks (GNNs) have gained increasing traction in the machine learning community. Graphs have long been used as a powerful abstraction for a wide variety of real-world data where structure plays a key role, from collaborations (Lima et al., 2014; Kipf & Welling, 2017) to biological (Gilmer et al., 2017; Ye et al., 2015) and physical (Shlomi et al., 2020) data, to mention a few. Before the advent of GNNs, *graph kernels* provided a principled way to deal with graph data in the traditional machine learning setting (Shervashidze et al., 2011; Bai et al., 2015; Minello et al., 2019). However, with the attention of machine learning researchers steadily shifting away from hand-crafted features toward end-to-end models where both the features and the model are learned together, neural networks have quickly overtaken kernels as the framework of choice to deal with graph data, leading to multiple popular architectures such as (Kipf & Welling, 2017; Gilmer et al., 2017; Veličković et al., 2018)

The main obstacle that both traditional and deep learning methods have to overcome when dealing with graph data is also the source of interest for using graphs as data representations. The richness of graphs means that there is no obvious way to embed them into a vector space, a necessary step when the learning method expects a vector input. One reason is the lack of a canonical ordering of the nodes in a graph, requiring either permutation-invariant operations or an alignment to a reference structure. Moreover, even if the order or correspondence can be established, the dimension of the embedding space may vary, as a result of structural modifications, i.e., changes in the number of nodes and edges.

In traditional machine learning, kernel methods, and particularly graph kernels, provide an elegant way to sidestep this issue by replacing explicit vector representations of the data points with a positive semi-definite matrix of their inner products. Thus, any algorithm that can be formulated in terms of scalar products between input vectors, can be applied to a set of data (such as graphs) on which a kernel is defined.

GNNs apply a form of generalised convolution operation to graphs, which can be seen as a message passing strategy where the node features are propagated over the graph to capture node interactions. In this paper we propose a neural architecture that bridges the two worlds of graph kernels and GNNs. The main idea is an analogy between the traditional convolution operator, which can be

seen as performing an inner product between an input matrix and a filter, and graph kernels, which compute an inner product of graphs. As such, graph kernels provide the natural tool to generalise the concept of convolution to the graph domain, which we term *graph kernel convolution* (GKC). Given a graph kernel of choice, in each GKC layer the input graph is compared against a series of structural masks (analogous to the convolutional masks in CNNs), which effectively represent learnable sub-graphs. These in turn can offer better intrepretability, in the form of insights into what structural patterns in the input graphs are related to the corresponding output.

**Our main contributions are:** 1. Our model is fully structural, unlike existing approaches that require embedding the input graph into a larger, relaxed space. While the latter allows one to seek more complex and arbitrary decision boundaries, it also has the potential of overfitting the problem and creating more local optima, as evidenced by the need to use dropout strategies to avoid overfitting in Graph Convolutional Networks (GCNs) (Rong et al., 2019; Cong et al., 2021). 2. Our architecture allows to plug-in any number and type of graph kernel (not just a single differentiable kernel as in Nikolentzos & Vazirgiannis (2020)). 3. As an added benefit, our model provides some interpretability in terms of the structural masks that are learned during the training process, similarly to convolutional masks in traditional CNNs. 4. Finally, we analyse the expressive power of our model and we show that it is greater than that of standard message-passing GNNs and the equivalent Weisfeiler–Lehman (WL) graph isomorphism test.

The fundamental goal of our model is to reconstruct the structural information needed for classification and is thus particularly suited for problems where the structure plays a pivotal role. These are often graph classification problems, where structure provides the most relevant information for classification, while node/edge features take a secondary role. The situation is usually reversed for node classification problems, where the distribution of features across the immediate neighbours of a node holds most of the information needed to classify the node itself. For these reasons in this paper we focus on graph classification problems. However we stress that our model can be indeed extended to tackle node-level tasks.

The remainder of this paper is organised as follows. In Section 2, we review the related work. In Section 3, we present our model, where graph kernels are used to redefine convolution on graph data. In Section 4, we investigate the hyper-parameters of our architecture through an extensive ablation study, provide some insight on the interpretability of the structural masks, and evaluate our architecture on standard graph classification benchmarks. Finally, Section 5 concludes the paper.

## 2 RELATED WORK

The majority of graph kernels belongs to one of two main categories: 1) bag-of-structures and 2) information propagation kernels. Bag-of-structures kernels compute the similarity between a pair of input graphs by first decomposing them into simpler sub-structures and then counting the number of isomorphic sub-structures between the two input graphs. Depending on the type of sub-structure considered, one can build a multitude of different kernels, e.g., sub-trees (Ramon & Gärtner, 2003), shortest paths (Borgwardt & Kriegel, 2005), and graphlets (Shervashidze et al., 2009). Information propagation kernels, on the other hand, include methods where pairs of input graphs are compared based on how information diffuses on them. Examples include random walk kernels (Kashima et al., 2003; Bai & Hancock, 2013), quantum walk kernels (Bai et al., 2015; Rossi et al., 2015), and kernels based on iterative label refinements (Shervashidze et al., 2011). While some kernels work only on undirected and unattributed graphs, other kernels are designed to handle attributes as well, either discrete- or continuous-valued (Shervashidze et al., 2011; Da San Martino et al., 2017). For a detailed review and historical perspective on graph kernels, we refer the reader to the recent survey of Kriege et al. (2020).

In recent years, with the advent of deep learning and the renewed interest in neural architectures, the focus of graph-based machine learning researchers has quickly moved to extending deep learning approaches to deal with graph data. Fundamentally, the principle underpinning most GNNs is that of exploiting the structure of the graph to propagate the node feature information iteratively. One of the first papers to propose the idea of GNNs is that of Scarselli et al. (2008), where an information diffusion mechanism is used to learn the nodes' latent representations by exchanging neighbourhood information. Depending on the form this diffusion takes, Bronstein et al. (2021) distinguish between convolutional (Kipf & Welling, 2017; Atwood & Towsley, 2016; Levie et al., 2018), atten-

tional Veličković et al. (2018), or message passing (Gilmer et al., 2017) types of GNNs, with the latter being the most general and formally equivalent to the Weisfeiler-Lehman graph isomorphism test under some technical conditions (Xu et al., 2018; Morris et al., 2019). More recently, Eliasof et al. (2021) proposed a reinterpretation of graph convolution in terms of partial differential equations on graphs, noting that different problems can benefit from different networks dynamics, hence suggesting the need to look beyond diffusion. For a comprehensive survey of GNNs we refer the reader to Wu et al. (2020).

In this work, we argue that a natural extension of the convolution operation to the graph domain, and thus an ideal candidate to build graph CNNs, already exists in the form of graph kernels. A number of works in the literature have investigated potential synergies between GNNs and kernels. Lei et al. (2017), introduce a class of deep recurrent neural architectures show that this lies in the reproducing kernel Hilbert space (RKHS) of graph kernels. Nikolentzos et al. (2018) compute continuous embeddings of graphs using kernels and plug them into a neural network. Xu et al. (2018); Morris et al. (2019) show that GNNs have the same expressiveness as the WL graph kernel (Shervashidze et al., 2011) and propose a new generalised architecture with increased expressive power. Du et al. (2019) take a somewhat different approach and exploit neural networks to introduce a new graph kernel kernel. This in turn is shown to be equivalent to an infinitely-wide GNN initialized with random weights and trained with gradient descent. Chen et al. (2020) propose a graph neural architecture where each layer enumerates local sub-structures around each node and then maps them to a RKHS via a Gaussian kernel mapping. In the context of graph compression, Bouritsas et al. (2021) learn how to best decompose the graph into small substructures that are also learned.

The closest work to ours we are aware of is that of Nikolentzos & Vazirgiannis (2020), who propose a GNN where the first layer consists of a series of hidden graphs that are compared against the input graph using a random walk kernel. However, due to their optimisation strategy, their model only works with a single differentiable kernel, whereas our model allows us to tap into the expressive power of any type and number of kernels. Moreover, in the architecture of Nikolentzos & Vazirgiannis (2020) the learned structural masks are assumed to be complete weighted graphs, whereas we allow for graphs with arbitrary structure.

## 3 Graph Neural Networks from Graph Kernels

### 3.1 Preliminaries

**Kernel methods** are a class of algorithms that are capable of learning when presented with a particular pairwise similarity measure on the input data, known as a kernel. Consider a set $X$ and a positive semi-definite kernel function $\mathcal{K} : X \times X \to \mathbb{R}$ such that there exists a map $\phi : X \to H$ into a Hilbert space $H$ and $\mathcal{K}(x, y) = \phi(x)^\top \phi(y)$ for all $x, y \in X$. Crucially, $X$ can represent any set of data on which a kernel can be defined, from $\mathbb{R}^d$ to a finite set of graphs. Hence, the field of machine learning is ripe with examples of graph kernels (see Section 2), which are nothing but positive semi-definite pairwise similarity measures on graphs. These can be either implicit (only $\mathcal{K}$ is computed) or explicit ($\phi$ is also computed).

**Weisfeiler-Lehman test.** The WL kernel (Shervashidze et al., 2011) is one of the most powerful and commonly used graph kernels and it is based on the 1-dimensional Weisfeiler-Lehman (1-WL) graph isomorphism test. The idea underpinning the test is that of partitioning the node set by iteratively propagating the node labels between adjacent nodes. With each iteration, the set of labels accumulated at each node of the graph is then mapped to a new label through a hash function. This procedure is repeated until the cardinality of the set of labels stops growing. Two graphs can then be compared in terms of their label sets at convergence, with two graphs being isomorphic only if their label sets coincide.

**Learning on graphs.** Let $\mathcal{G} = (V, X, E)$ be an undirected graph with $|V|$ nodes and $|E|$ edges, where each node $v$ is associated to a label $x(v)$ belonging to a dictionary $\mathbf{D}$. A common goal in graph machine learning problems is to produce a vector representation of $\mathcal{G}$ that is aware of both the node labels and the structural information of $\mathcal{G}$. Generalizing the convolution operator to graphs, GNNs learn the parameters $\Theta$ of a function $h$ that embeds the graph into a vector $Z$ by performing

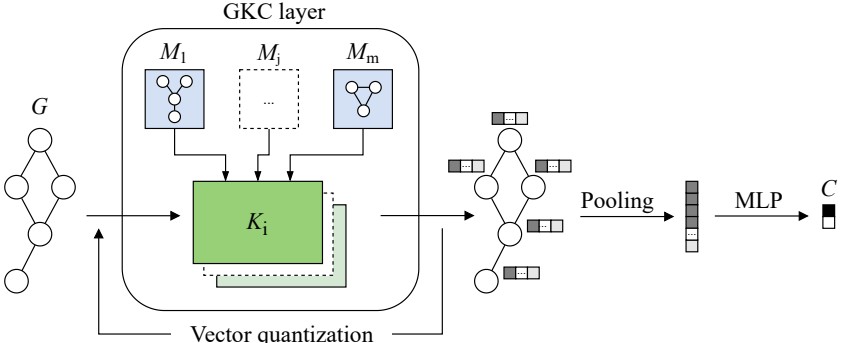

Figure 1: The proposed GKNN architecture. The input graph is fed into one or more GKC layers, where sub-graphs centered at each node are compared to a series of structural masks through a kernel function. The output is a new set of real-valued feature vectors associated to the graph nodes, which goes through a vector quantization operation. We obtain a graph-level feature vector through pooling on the nodes features, which is then fed to an MLP to output the final classification label.

message passing (Gilmer et al., 2017) in the 1-hop neighborhood $\mathcal{N}_{\mathcal{G}}^1(v)$ of each node $v \in V$, i.e.,

$$z(v) = \sum_{u \in \mathcal{N}_{\mathcal{G}}^1(v)} h_{\Theta}\left(x(v), x(u)\right). \tag{1}$$

The convolution is usually repeated for $L$ layers, and the final vectorial representation $Z$ is obtained by applying a node-wise permutation invariant aggregation operator $\square$ on the node features $z(v)$.

### 3.2 Proposed architecture

The core feature of the proposed model is the definition of a new approach to perform the convolution operation on graphs. As opposed to the diffusion process performed by message passing techniques, we design the *Graph Kernel Convolution* (GKC) operation in terms of the inner product between graphs computed through a graph kernel function.

Given a graph $\mathcal{G} = (V, X, E)$ with $n$ nodes, we extract $n$ sub-graphs $\mathcal{N}_{\mathcal{G}}^r(v)$ of radius $r$, centered at each vertex $v \in V$. Each sub-graph consists of the central node $v$, the nodes at distance at most $r$ from $v$, and the edges between them. Each such sub-graph is then compared to a set of structural masks $\{\mathcal{M}_1, \ldots, \mathcal{M}_m\}$ through

$$x_i(v) = \mathcal{K}(\mathcal{M}_i, \mathcal{N}_{\mathcal{G}}^r(v)), \tag{2}$$

where the resulting feature vector $x(v)$ is a non-negative real valued $m$-dimensional vector collecting the kernel responses, and the $i$th structural mask $\mathcal{M}_i = (L_{\mathcal{M}_i}, E_{\mathcal{M}_i})$ is a graph with node set $L_{\mathcal{M}_i}$ (including node labels) and edge set $E_{\mathcal{M}_i}$.

Since in this paper we focus on graph classification, the final part of our architecture, as depicted in Figure 1, consists of applying a node-wise pooling operation on the previously computed feature vectors to obtain a global graph descriptor. This is then fed to a multilayer perceptron (MLP) layer to obtain the final classification.

### 3.3 Multi-layer architecture

Most graph kernels, including the WL kernel, work with discrete node labels. In our GKNN architecture, however, equation 2 yields an $m$-dimensional real valued vector for each graph node. Hence, we need to discretize the output feature space before being able to perform a further convolution operation. Before each convolution layer, except for the first one (unless the input node features are continuous), we add a *vector quantization* operation to the input node features, whereby the feature space is discretized using $k$-means clustering on the node features of the current batch. Nodes are then labeled according to the cluster index they belong to. To keep the labeling consistent

among different mini-batches, the $k$-means algorithm centroids at step $t+1$ are initialized using the centroids computed at the previous step $t$, i.e.,

$$x_i^{t+1}(v) = \mathcal{K}(\mathcal{M}_i, \mathcal{N}_{\mathcal{G}(V, vq(X^t), E)}^r(v)), \tag{3}$$

where $vq(X)$ indicates the vector quantization operation described above[1] .

Note that in our current implementation we only compute the gradient of the kernel value with respect to the masks of the current layer, and not with respect to the input graph for that layer, which would be required to backpropagate the gradient to the previous layer. This is due to the presence of the vector quantization layer which complicates the backpropagation with respect to node features. To overcome this limitation and allow the optimization of the structural masks in each layer, we add skip connections between the layers and masks weights are updated only along these paths. Empirically we observed this not to be a major problem as longer paths have diminishing weight. The final graph embedding after $L$ layers is thus obtained by $\square(Z^1 | \ldots | Z^L)$, where $Z^\ell$ are the output features of the $\ell$th layer and $\cdot | \cdot$ indicates the concatenation of node-wise features. In particular, in all our experiments we use sum pooling as the aggregation operator $\square$.

### 3.4 Optimization strategy

Our learning problem can be formulated as follows: given a set $\{\mathcal{G}_1, \ldots, \mathcal{G}_B\}$ of $B$ training input graphs with node labels belonging to the dictionary $\mathbf{D}$ and associated class labels $y_1 \ldots y_B$, our goal is to find the optimal parameters ($\Phi$ and masks $\mathcal{M}_i$ ) of the model $g$ for the following minimization problem,

$$\min_{\mathcal{M}_1 \ldots \mathcal{M}_m, \Phi} \sum_{i=1}^{B} CrossEntropy(g_{\mathcal{M}_1 \ldots \mathcal{M}_m, \Phi}(\mathcal{G}_i), y_i). \tag{4}$$

There are two main challenges that we need to consider when optimizing the structural masks. First, the number of nodes of the sub-graphs is not fixed and can in principle vary from $1$ (for isolated nodes) to $n$ (for fully connected graphs). Second, since graph kernel functions are not in general differentiable, the automatic differentiation mechanism of common neural network optimization libraries cannot be directly applied to our model.

**Structural kernels representation**   When defining the space of graphs on which we want to optimize the structural masks $\mathcal{M}$, we have to consider the possible sub-structures present in the input graph that characterize it as belonging to a specific class. Assuming that this knowledge is not known *a priori*, we should allow to learn structures as large as the graph itself. Unfortunately, since the space of graphs grows exponentially with the number of nodes, this would be impractical. Moreover, under the assumption of the presence of localized characterizing substructures, we usually need graphs of few nodes to capture their presence. In our implementation, we fix a maximum number of nodes $d$ for each substructure and optimize for structural masks in the space

$$\mathcal{M} \in \bigcup_{p=1}^{d} \left( \tilde{\mathbf{G}}_p \times \mathbf{D}^p \right), \tag{5}$$

where $\tilde{\mathbf{G}}_p$ indicates the set of all possible connected graphs of $p$ nodes, and $\mathbf{D}^p$ the labeling space of $p$ nodes. The impact of this hyper-parameter is studied in the ablation study in Section 4.

**Structural kernels optimization**   Graph kernels, as functions operating on discrete graph structures, are in general not differentiable. In order to be able to optimize the structural masks we use a Discrete Randomized Descent (DRD) strategy. Given an initial structural mask $\mathcal{M}$, we let the graph evolve to $\mathcal{M}'$ through edit operations minimizing a cost function. These operations consist of adding or removing some edges, and changing node labels. After the editing operation we extract the maximum connected component in $\mathcal{M}'$, thus allowing us to consider all the graphs with nodes $p \leq d$ without explicitly optimizing over $p$.

---

[1]We verified the displacement of the centroids induced by the k-means clustering in different models and datasets. With a batch size of 16, at convergence, the displacement is always within 0.9% of the average distance between the centroids.

The DRD update is performed during the backpropagation phase of the model training step. Through the chain rule mechanism we can estimate the discrete gradient of the final classification loss w.r.t. the structural mask $\mathcal{M}$ as $\frac{\delta loss}{\delta \mathcal{M}} = \sum_v \frac{\delta loss}{\delta x(v)} \cdot \frac{\delta x(v)}{\delta \mathcal{M}}$. The discrete sub-gradient of the kernel response $x(v)$ w.r.t. $\mathcal{M}$ *i.e., the gradient restricted to the randomly selected edit operations,* can be estimated as the difference between the kernel responses after and before an edit operation:

$$\frac{\delta loss}{\delta \mathcal{M}} = \sum_v \frac{\delta loss}{\delta x(v)} \cdot \left( \mathcal{K}(\mathcal{M}', \mathcal{N}_G^r(v)) - \mathcal{K}(\mathcal{M}, \mathcal{N}_G^r(v)) \right). \tag{6}$$

During each backpropagation step, we thus sample an edit operation $e$ from the set of all edit operations on $E_\mathcal{M}$, and accept the edit only if the value of equation 6 is lower than or equal to 0. This means that we always evolve toward a graph locally minimizing the loss function.

Together with the structural mask $\mathcal{M}$, we also optimize a probability distribution over the edit operations. The derivative of an edit operation probability $p(e)$ is estimated using the same equation 6 and optimized with standard gradient descent optimizers. Further details are in the Appendix A.1.

**Jensen–Shannon divergence loss**   Depending on the number of structural masks to learn, we experimentally observed that different structural masks can give a similar response over the same node. To avoid this behavior and push the model to learn a more descriptive node feature vector, we propose to regularize the structural masks learning process by adding a Jensen–Shannon divergence (JSD) loss. The JSD is computed between the feature dimensions considered as probability distributions over the nodes of the graph.

Let $P_i = \{\alpha x(v)_i | v \in V\}$ be the probability distribution induced by the $i$th kernel over the graph nodes. $\alpha$ is a scaling factor ensuring that $\sum_v P_i(v) = 1$. We define the JSD loss as

$$loss_{JSD} = -H \left( \sum_{i=1}^n P_i \right) + \sum_{i=1}^n H\left(P_i\right), \tag{7}$$

where $H(P)$ is the Shannon entropy. The final loss we optimize is then the sum of 1) the CrossEntropy loss defined in equation 3.4 and 2) $loss_{JSD}$ multiplied by a weighting factor.

## 3.5   EXPRESSIVE POWER

In this subsection, we study the expressive power of the proposed architecture. Recall from Morris et al. (2019) that the expressive power of standard GNNs (i.e., GNNs that only consider the immediate neighbours of a node when updating the labels) is equivalent to that of the 1-WL test. We argue that GKNN has a higher expressive power than the 1-WL test (and thus standard message-passing GNNs). To show this, we start by demonstrating that every pair of graphs that can be distinguished by the 1-WL test can also be distinguished by our model. Without loss of generality, in the following we consider two unlabelled graphs $\mathcal{G}_1 = (V_1, E_1)$ and $\mathcal{G}_2 = (V_2, E_2)$ with the same number of nodes $n = |V_1| = |V_2|$.

**Theorem 1.** *Given two input graphs $\mathcal{G}_1$ and $\mathcal{G}_2$, if the 1-WL test can distinguish between them then there exists an instance of a GKNN that can also distinguish them.*

A sketch of proof of Theorem 1 can be found in Appendix A.2. Next, we show that the GKNN can distinguish between pairs of graphs where the 1-WL test fails. To see that this is the case, consider the standard example of two graphs with 6 nodes, where the first graph is made of two disconnected cycles with 3 nodes, while the second graph is a single cycle with 6 nodes. Both graphs are regular, with every node of the graphs being adjacent to two other nodes, i.e., every node has degree 2. As explained in Nikolentzos et al. (2020), despite being non-isomorphic, the two graphs are considered identical by the 1-WL test (in other words, the WL relabeling procedure converges to the same set of labels for the two graphs). To see why this happens, consider that through the propagation phases each node is only made aware of what degree other nodes in the graph have and not how many such nodes can be reached. As the following lemma shows, this is not an issue for the GKNN:

**Lemma 2.** *The GKNN is able to identify graph containing triangles.*

A sketch of proof of Lemma 2 can be found in Appendix A.2. From Theorem 1 and Lemma 2 it follows that the GKNN has a higher expressive power than the 1-WL test and thus standard GNNs.

Figure 2: Ablation study: bar plots of classification accuracy with standard error. Left to right: number of nodes (i.e., structural mask size), number of structural masks, kernel functions (WL, WL with Optimal Assignment, Graphlet, Propagation, and Pyramid match kernel), subgraph radius, number of GKC layers, and weight of the JSD loss.

# 4 EXPERIMENTAL EVALUATION

## 4.1 SENSITIVITY STUDY

We performed an ablation study to investigate how the various components and hyper-parameters of our architecture affect the model performance. To this end, we consider the MUTAG dataset (for more details on the datasets, see Appendix A.7). We start by setting a baseline network configuration (one layer, 16 structural masks of 6 nodes each, subgraphs radius of 3, WL graph kernel and JSD weight of $10^{-4}$) and then we let the considered hyper-parameter(s) vary. Specifically, we study the influence of the number of nodes (i.e., the maximum size of the structural masks), number of structural masks, kernel function, subgraph radius, number of GKC layers, and weight of $loss_{JSD}$.

We report accuracy results as bar plots with standard error in Figure 2 (from left to right: number of nodes, number of structural masks, kernel function, subgraph radius, number of GKC layers, and weight of the JSD loss). Both the number of structural masks and the maximum number of their nodes play a crucial role in the classification accuracy and thus need to be carefully chosen. As expected, also the choice of the kernel function is a crucial factor, highlighting the advantage of allowing to plug-in any non-differentiable graph kernel function. As for the subgraph radius, the number of neighbors clearly influences the model performance and there seem to be an optimal subgraph size able to catch the structural characteristics of the input graphs. The ablation validates also our multilayer architechture design, even if the model requires just two layers to reach the best accuracy. This is justified by the wider neighborhood involved in our convolution operation compared to the standard message passing formulation. The results also confirm the importance of the JSD loss, as the performance of the GKNN increases once the loss is introduced.

## 4.2 INTERPRETABILITY ANALYSIS

One of the factors that fostered interpretability in classical CNNs is the possibility to visualize and analyze the learned filters capturing the fundamental structures characterizing the input images (Nikolentzos & Vazirgiannis, 2020). This feature is missing in the classical formulation of graph convolution relying on the message passing paradigm. On the other hand, our method learns actual graph masks, potentially increasing the interpretability of the model by allowing us to probe into the explanatory factors of variation behind the input data through the learned structural masks.

To show the potential of our model to capture fundamental structures in the input graphs, we devised a simple but effective synthetic experiment. First, we train our model on a binary classification task, i.e., predicting if a certain graph motif is present or not in the given graph. Then we qualitatively assess if the learned structural masks have captured or not the graph motif. Following the setup of Nikolentzos & Vazirgiannis (2020), we created 5 different datasets, one for each of the graph motif depicted in Figure 3 (upper row). A detailed description of the synthetic datasets and training details can be found in the Appendix A.6. The model is then trained end-to-end to predict the graph labels on a 90/10% train/test split.

The results show that the learned structural masks have very similar structures to those of the corresponding motifs, with only a few missing or misplaced edges (see for instance the columns corresponding to the ring and the wheel). Moreover, the distribution of the responses in the original graphs clearly highlights the motifs position. Overall, the results demonstrate that our model is able to extract the salient structural patterns, which in turn allows us to understand what features were

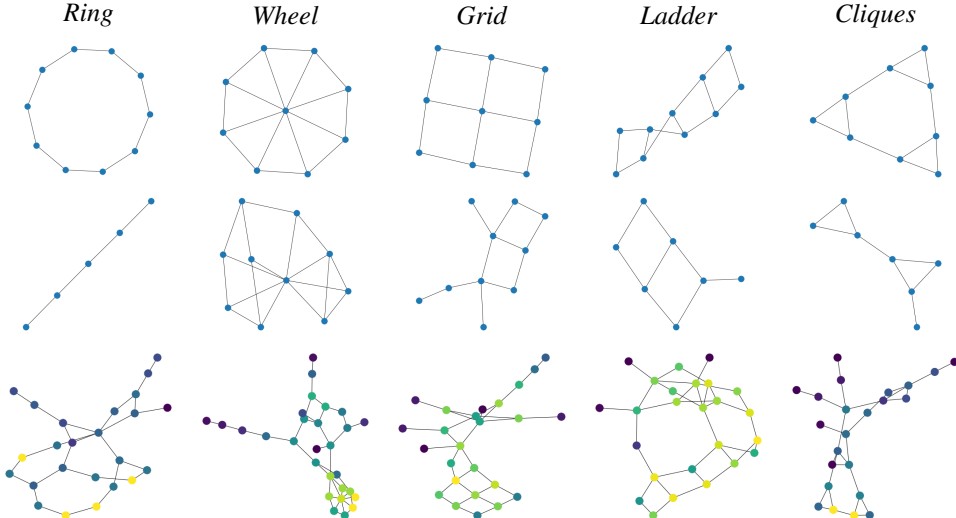

Figure 3: Interpretability analysis results. Each column refers to a graph motif. Upper row: original motif. Mid row: structural mask with the strongest response. Bottom row: a sample graph including the motif. Colors indicate the node response to the filter, with lighter colours (yellow) indicating a high response and darker colors (blue) indicating a low response.

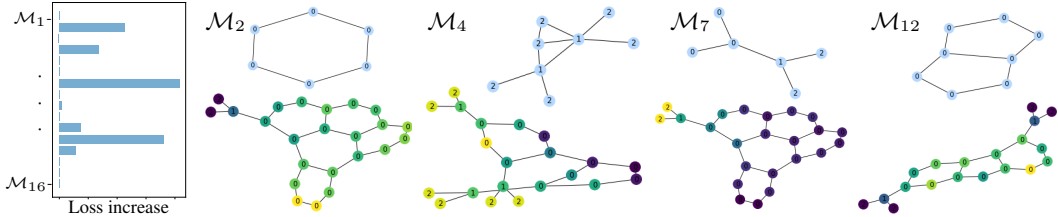

Figure 4: Top 4 significant structural masks (top) learned by our model on the MUTAG dataset and their response over input graph nodes (bottom). The left bar plot shows the impact of each structural mask on the classification loss.

detected for a given input graph. We would like to stress that even though this aspect is not the final goal of this work, the interpretability of our model definitely helps to foster trust in our approach.

To further investigate the ability of our model to capture salient structural features, we show in Figure 4 some of the structural masks learned on the MUTAG dataset. In particular, we trained a model with the hyper-parameters of the baseline architecture used in the ablation study of Section 4.1. To select the most significant filters, we manually set to zero the response of the $i$th filter $\mathcal{M}_i$ and evaluate again the classification loss. In the bar plot (Figure 4, left) we report the loss increase after zeroing each filter response. Below each mask we also show the input graph of the training set that gave the highest kernel response among all nodes together with the per-node response as node colors (yellow indicating a high response, dark blue indicating a low response).

## 4.3 GRAPH CLASSIFICATION RESULTS

**Datasets** We evaluate the performance of the proposed model on the graph classification task. We make use of publicly available datasets frequently employed in the GNNs literature (Kersting et al., 2016). In particular, we use 4 bio/chemo-informatics datasets collecting molecular graphs (MUTAG, NCI1, PROTEINS, and PTC) and a social dataset (IMDB-BINARY). Bio/chemo-informatics graphs differ from social graphs as in the former nodes have categorical input features, whereas in the latter there are no features. More details on these datasets can be found in Appendix A.7.

**Experimental setup** We compare our model against: • 5 state-of-the-art GNNs: DGCNN (Zhang et al., 2018), DiffPool (Ying et al., 2018), GIN (Xu et al., 2018), and (s)GIN (Di et al., 2020); • two

Table 1: Classification results on chemo/bio-informatics and social datasets. Mean accuracy and standard error are reported. The best performance (per dataset) is highlighted in bold.

| | MUTAG | PTC | NCI1 | PROTEINS | IMDB |
|---|---|---|---|---|---|
| Baseline | 78.57 ± 4.00 | 58.34 ± 2.02 | 68.50 ± 0.87 | 73.05 ± 0.90 | 49.50 ± 0.79 |
| WL | 82.67 ± 2.22 | 55.39 ± 1.27 | **79.32 ± 1.48** | 74.16 ± 0.38 | **71.80 ± 1.03** |
| DiffPool | 81.35 ± 1.86 | 55.87 ± 2.73 | 75.72 ± 0.79 | 73.13 ± 1.49 | 67.80 ± 1.44 |
| GIN | 78.13 ± 2.88 | 56.72 ± 2.66 | 78.63 ± 0.82 | 70.98 ± 1.61 | 71.10 ± 1.65 |
| DGCNN | 85.06 ± 2.50 | 53.50 ± 2.71 | 76.56 ± 0.93 | 74.31 ± 1.03 | 53.00 ± 1.32 |
| GraphSAGE | 77.57 ± 4.22 | **59.87 ± 1.91** | 75.89 ± 0.96 | 73.11 ± 1.27 | 68.80 ± 2.26 |
| sGIN | 84.09 ± 1.72 | 56.37 ± 2.28 | 77.54 ± 1.00 | 73.59 ± 1.47 | 71.30 ± 1.75 |
| RWGNN | 82.51 ± 2.47 | 55.47 ± 2.70 | 72.94 ± 1.16 | 73.95 ± 1.32 | 69.90 ± 1.32 |
| **Ours** | **85.73 ± 2.70** | 58.39 ± 3.40 | 71.52 ± 1.12 | **74.48 ± 1.10** | 69.70 ± 2.20 |

distinct baselines, depending on the dataset type (see below): Molecular Fingerprint (Ralaivola et al., 2005; Luzhnica et al., 2019) and Deep Multisets (Zaheer et al., 2017); • the WL kernel (Shervashidze et al., 2011); • RWNN, a GNN model employing a differentiable graph kernel (Nikolentzos & Vazirgiannis, 2020), the closest existing neural architecture to ours. For the WL subtree kernel we use a $C$-SVM (Chang & Lin, 2011) classifier. For the baselines, we follow Errica et al. (2020) and use the Molecular Fingerprint technique (Ralaivola et al., 2005; Luzhnica et al., 2019) for chemical datasets. For the social dataset we rely on the permutation invariant model of Zaheer et al. (2017).

To ensure a fair comparison, for each of these methods we follow the same experimental protocol. We perform 10-fold cross validation where in each fold the training set is further subdivided in training and validation with a ratio of 9:1. The validation set is used for both early stopping and to select the best model within each fold. Importantly, folds and train/validation/test splits are consistent among all the methods. For all the methods we perform grid search to optimize the hyperparameters. In particular, for the WL method we optimize the value of $C$ and the number of WL iterations $h \in \{4, 5, 6, 7\}$. For the RWGNN we investigate the hyper-parameter ranges used by the authors (Nikolentzos & Vazirgiannis, 2020), while for all the other GNNs we follow Errica et al. (2020) and we perform a full search over the hyper-parameters grid. For our model, we explore the following hyper-parameters: number of structural masks in {8,16,32}, maximum number of nodes of a structural mask in {6,8}, subgraph radius in {1,2,3}, number of layers in {1,2,3} and always use WL as graph kernel. In all our experiments, we train the model for 1000 epochs, using the Adam optimizer with learning rate of 0.001 for MLP weights and of 0.01 for the edit operation probabilities, and a batch size of 32. The MLP takes as input the sum pooling of the node features and is composed by two layers of output dimension $m$ and $c$ (# classes) with a ReLU activation in between.

**Results** The accuracy for each method and dataset is reported in Table 1. For the bio/chemo-informatics datasets, the performance of our approach is on par with the best ones, as we can see for MUTAG, PROTEINS, and PTC. This holds as long as the number of node labels is low. Indeed, in the case of high-dimensional node labels (e.g., NCI1), the classification accuracy tends to decrease as the optimization becomes harder. As for the social dataset, the performance of our approach is satisfactory, particularly considering the lack of discriminative substructures in social networks.

## 5 CONCLUSION

We introduced a new convolution operation on graphs based on graph kernels. We proposed the graph kernel convolution layer and an architecture that makes use of this layer for graph classification purposes. The benefits of this architecture include the definition of a fully structural model that can exploit the vast collection of existing graph kernels, a provably superior expressive power when compared to standard GNNs, and the possibility to visualize the learned substructures in a way that is reminiscent of the convolutional filters of standard CNNs. Future work will attempt to address the limitations of the current approach. For example, the optimization strategy we employed in this work prefers low-dimensional discrete input labels over high-dimensional continuous ones. Our model is also not well suited for social graphs, where structure plays a minor role and small-worldness implies that the sub-graphs will span the majority of the input graph even at small radii. Finally, the lack of widely available GPU implementations of graph kernels implies that our model cannot presently tap into the computational power of these units.

## REPRODUCIBILITY STATEMENT

To maximize reproducibility of our results by the academic community, we provide an extensive discussion of the model, its hyper-parameters, the training and the experimental setup. We also make use of widely used publicly available datasets. Finally, we include the source code in the supplementary material and we will make it publicly available upon acceptance.

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

# A APPENDIX

## A.1 EDIT OPERATIONS PROBABILITY OPTIMIZATION

Together with the structural mask $\mathcal{M}$, we optimize also the probabilities $p(e_{i,j}) \forall i, j = 1 \dots p$ of each edge to belong to the structural mask, and the probability $p(l_i = d) \forall i = 1 \dots p, d \in D$ of each node $i$ to assume the label value $d$. Given a structural mask $\mathcal{M} = (L_\mathcal{M}, E_\mathcal{M})$ with node labels $L = l_1 \dots l_p$ and edge set $E$, the probability distribution of edit operations can be derived from the previous edge and label probabilities as

$$p(E' = E \cup e_{ij}) = \begin{cases} 0 & \text{if } e_{ij} \in E \\ \frac{p(e_{ij})}{\sum_{e_{s,t} \in E} p(e_{s,t}) + \sum_{e_{s,t} \notin E}(1 - p(e_{s,t}))} & \text{otherwise} \end{cases} \quad (8)$$

$$p(E' = E - e_{ij}) = \begin{cases} 0 & \text{if } e_{ij} \notin E \\ \frac{1 - p(e_{ij})}{\sum_{e_{s,t} \in E} p(e_{s,t}) + \sum_{e_{s,t} \notin E}(1 - p(e_{s,t}))} & \text{otherwise} \end{cases} \quad (9)$$

for edge edit operations consisting in either add or remove and edge $e_{i,j}$, while for changing the label $l_i$ of the $i$th node to $d$ as

$$p(l_i' = d, l_j' = l_j \forall i \neq j) = \begin{cases} 0 & \text{if } l_i = d \\ \frac{p(l_i = d)}{\sum_{j, \hat{d} \in D, l_j \neq d} p(l_j = d)} & \text{otherwise} \end{cases} \quad (10)$$

To keep the optimization simpler, we keep two separate distributions for edge and node label editing operations and alternate between the two. The derivatives of edge and node label probabilities are computed according to equation 6 and used as input to standard gradient descent based optimizers (e.g., Adam).

## A.2 EXPRESSIVE POWER

**Sketch of proof of Theorem 1** This is trivially the case when we adopt the WL kernel in the GCK layer and the radius $r$ is such that the sub-graphs $N_{\mathcal{G}_1}^r(v)$ span the entire graph $\mathcal{G}_1$ (similarly for $\mathcal{G}_2$). Under this assumption, the set of labels computed against any one mask will be the same for every node in graph $\mathcal{G}_1$ ($\mathcal{G}_2$), since all the sub-graphs are identically equal to the graph itself. Assuming that we have a set of structural masks $M = \{\mathcal{M}_1, \cdots, \mathcal{M}_m\}$ that span the space of graphs over $n$ nodes, then, under the hypothesis that the 1-WL test can distinguish between $\mathcal{G}_1$ and $\mathcal{G}_2$, the set of labels computed by the GCK layer (i.e., the responses of the masks) while identical on all the nodes of each graph, will differ between the two graphs. In particular, the response will peak for any mask fractionally equivalent to the graph itself (Ramana et al., 1994). Therefore, under any reasonable pooling strategy, the GKNN will be able to distinguish between the two graphs.

We now prove that the same holds for an arbitrary sub-graph radius $r$, under the assumption of a sufficient number of layers. For simplicity, we will assume $r = 1$, but as shown in the previous argument, increasing $r$ increases the descriptive power of the GCK layer. The prove derives from the fact that the full GCK Layer can be seen as 1) a label propagation step followed by 2) a hashing step. The label propagation is given by the computation of the WL kernel against a structural mask, due to the 1-WL iterations on the sub-graphs.

The hashing, on the other hand, comes from the implicit dot product against the structural masks. This is in essence a projection onto the space spanned by them and forces a hashing where the adopted masks form a representation space for the hashing itself. Note that, under the assumption of a sufficiently large (possibly complete) set of structural masks $M = \{\mathcal{M}_1, \cdots, \mathcal{M}_m\}$ spanning the space of graphs over $n$ nodes, the dot product induced hashing will not introduce conflicts between nodes with different labels. In other words, labels that are distinct under the 1-WL iterations will eventually be mapped to distinct labels at some GCK layer. Note, however, that under these assumptions the expressive power is greater or equal than that of 1-WL and labels that are equivalent under 1-WL iteration can indeed be mapped to distinct labels by some GCK layer. In fact, the propagation for 1-WL uses only the star of radius 1 around a particular node, while our scheme uses the full 1-hop sub-graph, which includes connections between neighbors . Therefore, given a sufficient number of GCK layers and a reasonable pooling strategy, the GKNN will be able to distinguish between the two input graphs.

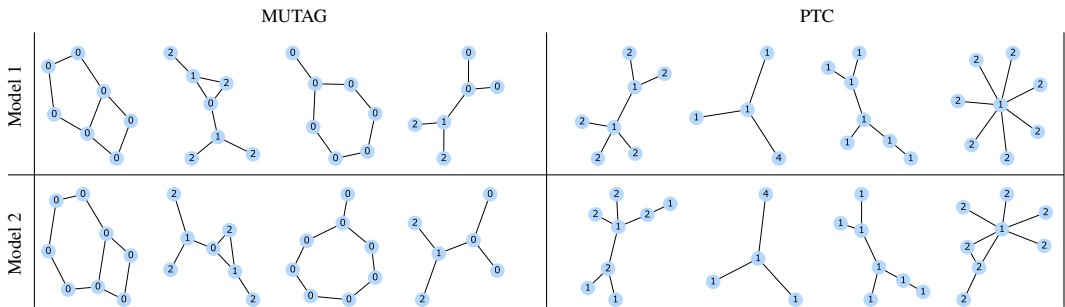

Figure 5: Top 4 significant structural masks learned by our model on the MUTAG (left) and PTC (right) datasets starting from two different random initializations of the model weights (rows).

**Sketch of proof of Lemma 2** Assume we have a GCK layer with radius $r = 1$ that adopts a WL kernel and a set of structural masks $M = \{\mathcal{M}_1, \cdots, \mathcal{M}_m\}$ representing all stars up to $n - 1$ nodes, where $n$ denotes the number of nodes of the input graph. The presence of a triangle in the graph induces the existence of at least three sub-graphs of radius 1 with connections between neighbors, i.e., not stars . On these sub-graphs, no star masks will achieve maximum similarity under the WL kernel. Having a pooling process that normalizes the masks responses w.r.t. size and maximizes over the masks on the same node, implies that each node will have the same maximal response if it does not contain triangles and a lower response otherwise. A min-pooling process over the node features will then be able to discriminate between graphs with and without triangles.

Let the radius of the sub-graphs in equation 2 be $r = 1$. Then the sub-graphs built around the nodes of $\mathcal{G}_1$ and $\mathcal{G}_2$ will be structurally different, i.e., triangles in $\mathcal{G}_1$ and 3 nodes path graphs in $\mathcal{G}_2$. As a consequence, the feature vectors $x(v)$ over the nodes of the two graphs will be different, allowing the GKNN to distinguish between them.

### A.3 CONVERGENCE ANALYSIS

To experimentally assess the convergence of our optimization strategy we investigated the behavior of the structural masks both at convergence and during the optimization. In Figure 5 we show the most significant (see 4.2 for the definition) structural masks learned by two different training instances of the same architecture. Both models end up learning similar structural masks, indicating that our optimization is able to reach a stable local optimum. This is also supported by the interpretability analysis of Section 4.2, showing that the learned masks resemble the local substructures of the input graphs.

In Figure 6 we also investigate the evolution of different initializations of the structural mask weights leading to the same final structure in two different models. As we can see, labels have a faster convergence than the connectivity, this is probably due to the bigger impact that they have on the similarity score computed by graph kernels (WL in this specific case). On the right we report also the decreasing curves of the JSD and cross-entropy losses during the training of a model on MUTAG together with the cross-entropy loss computed on the validation-set.

### A.4 COMPUTATIONAL COMPLEXITY AND RUNTIME ANALYSIS

The computational complexity of our model is linear with respect to the number of input graph nodes and polynomial with respect to the sub-graph radius $r$, where the exact complexity depends on the choice of kernel.

In terms of runtime, for the datasets considered in this paper, this translates to the results shown in Table 2, where we report the average training time per dataset (both total and per epoch) of the models trained during the grid search and the average inference time of the best model. Both training and testing of each model were performed using just a single core of an Intel(R) Xeon(R) CPU E5-4627 v3 @ 2.60GHz processor. Note that training and test speed of a single model could be significantly improved by parallelizing the computation over the batch samples or by using multithread implementations of graph kernels.

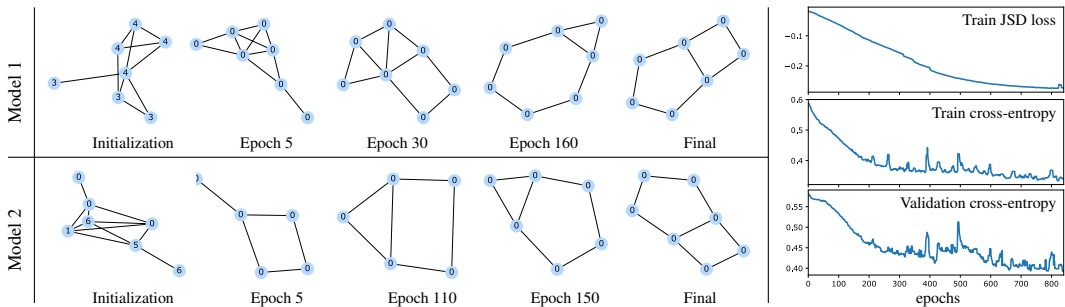

Figure 6: Left: Evolution of the learned structural masks during the training epochs showing two different initializations (belonging to two different models) that led to the same structural mask on the MUTAG dataset. Right: Sample of training curves of a model trained on MUTAG.

Table 2: Average training time (both total and per epoch) and inference time for the datasets considered in the present work. The two rightmost columns show some relevant datasets statistics that influence the running time. The standard deviation is reported with the symbol ±.

|  | Total training time (h) | Training time per epoch (s) | Test time per sample (s) | # Graphs | Avg # Nodes |
|---|---|---|---|---|---|
| PROTEINS | 12.27 ± 8.27 | 71.82 ± 53.60 | 0.10 ± 0.06 | 1113 | 39.06 |
| MUTAG | 1.43 ± 0.72 | 6.58 ± 3.35 | 0.11 ± 0.02 | 188 | 17.93 |
| PTC | 2.78 ± 2.83 | 22.61 ± 19.91 | 0.08 ± 0.06 | 344 | 14.69 |
| NCI1 | 21.35 ± 14.47 | 217.35±156.70 | 0.04 ± 0.01 | 4110 | 29.87 |
| IMDB | 6.65 ± 1.98 | 35.46 ± 7.89 | 0.11 ± 0.02 | 1000 | 19.77 |

## A.5 RANDOM GRID SEARCH TRAINING STRATEGY

In Section 4.3 we reported the results after a full search on the considered hyper-parameters grid. Although this is the most commonly adopted protocol for the considered datasets ( Errica et al. (2020); Nikolentzos & Vazirgiannis (2020); Xu et al. (2018)), in practical settings it may not be always feasible to explore the full hyper-parameters space. A straightforward strategy to reduce the number of models to test is through a random grid search, where the best model is determined based on a subset of randomly chosen hyper-parameter combinations. To assess the behavior of our architecture on this setting we report in Table 3 the mean and standard error of the average accuracy over all folds of 30 runs of random grid-search considering different numbers of tested models (5, 10, 20, 30, all). This is compared against the perfomance of GraphSAGE, sGIN, and RWNN.

Our model shows a similar behavior to that of most of our competitors, with a benefit when including more models in the evaluation. The average drop in performance from randomly selecting 5 models in contrast to performing a full grid search is around 2% and decreases with the number of models considered. The only exception seems to be sGIN, whose performance slightly decreases considering more models, probably due to a lower sensitivity of the model to the explored hyper-parameters, as suggested also by the lower standard error.

## A.6 SYNTHETIC EXPERIMENTS

**Datasets** For the interpretability analysis experiments we built a dataset for each of the 5 graph motifs. Each dataset is composed of 2,000 graphs, equally divided into positive and negative samples, with a varying number of nodes between 30 and 50. Each sample pair is generated starting from the same Erdős–Rényi graph, with an edge observation probability of 0.1. Next, for each Erdős–Rényi graph, two new graphs are built. The first one, labeled as 1, is created by connecting each node of the Erdős–Rényi to the nodes of the structural motif with probability 0.02. The second one, labeled as 0, is obtained in a similar manner, however instead of a motif we insert a random graph with the same number of nodes and edges of the motif structure.

Table 3: Mean accuracy and standard error when performing a random search with increasing number of tested models.

| | 5 | 10 | 20 | 30 | ALL |
|---|---|---|---|---|---|
| **MUTAG** | | | | | |
| GraphSAGE | 75.52 ± 0.62 | 75.54 ± 0.96 | 75.66 ± 1.12 | 74.48 ± 0.72 | 77.57 |
| sGIN | 84.56 ± 0.46 | 84.51 ± 0.38 | 84.45 ± 0.27 | 84.11 ± 0.13 | 84.09 |
| RWNN | 84.62 ± 0.59 | 84.22 ± 0.61 | 83.77 ± 0.64 | 83.87 ± 0.58 | 82.51 |
| Ours | 84.72 ± 0.66 | 85.15 ± 0.61 | 85.08 ± 0.64 | 84.81 ± 0.52 | 85.73 |
| **NCI1** | | | | | |
| GraphSAGE | 69.79 ± 0.98 | 68.67 ± 0.88 | 71.49 ± 0.74 | 73.49 ± 0.34 | 75.89 |
| sGIN | 76.68 ± 0.25 | 76.54 ± 0.29 | 77.05 ± 0.13 | 77.41 ± 0.11 | 77.54 |
| RWNN | 70.92 ± 0.68 | 72.11 ± 0.33 | 72.62 ± 0.32 | 73.02 ± 0.23 | 73.82 |
| Ours | 68.21 ± 0.25 | 68.45 ± 0.18 | 69.16 ± 0.22 | 69.51 ± 0.18 | 70.12 |
| **PTC** | | | | | |
| GraphSAGE | 58.22 ± 0.72 | 57.48 ± 0.67 | 58.21 ± 0.47 | 57.3 ± 0.74 | 59.87 |
| sGIN | 58.37 ± 0.58 | 57.3 ± 0.58 | 57.05 ± 0.42 | 56.63 ± 0.36 | 56.38 |
| RWNN | 57.69 ± 0.72 | 58.53 ± 0.52 | 57.60 ± 0.65 | 57.93 ± 0.61 | 55.47 |
| Ours | 56.87 ± 0.85 | 57.65 ± 0.60 | 57.74 ± 0.84 | 58.22 ± 0.51 | 58.41 |
| **PROTEINS** | | | | | |
| GraphSAGE | 70.86 ± 0.39 | 70.73 ± 0.42 | 71.62 ± 0.56 | 72.49 ± 0.19 | 74.31 |
| sGIN | 73.50 ± 0.27 | 73.89 ± 0.14 | 73.61 ± 0.14 | 73.80 ± 0.10 | 73.59 |
| RWNN | 72.80 ± 0.30 | 73.46 ± 0.15 | 73.50 ± 0.18 | 73.37 ± 0.14 | 73.95 |
| Ours | 74.36 ± 0.19 | 74.35 ± 0.19 | 74.37 ± 0.16 | 74.49 ± 0.11 | 74.48 |
| **IMDB** | | | | | |
| GraphSAGE | 55.90 ± 1.06 | 57.56 ± 1.00 | 59.96 ± 0.98 | 64.96 ± 0.35 | 68.80 |
| sGIN | 71.99 ± 0.20 | 71.90 ± 0.20 | 71.73 ± 0.19 | 71.48 ± 0.08 | 71.40 |
| RWNN | 69.63 ± 1.22 | 70.33 ± 0.30 | 69.87 ± 0.28 | 69.92 ± 0.29 | 69.90 |
| Ours | 66.94 ± 0.46 | 67.71 ± 0.41 | 68.72 ± 0.27 | 68.72 ± 0.27 | 68.70 |

**Training details** We trained the network with the same hyper-parameters as in the ablation baseline in section 4.1, with the exception of the number of structural masks, which is set to 8. Moreover, we replaced the ReLU activation of the final MLP with a Sigmoid activation.

## A.7 DATASETS

We validated our approach on a graph classification experiment on the following real-world datasets.

MUTAG (Debnath et al., 1991) is a dataset consisting of 188 mutagenic aromatic and heteroaromatic nitro compounds (with 7 discrete labels) assayed for mutagenicity in Salmonella typhimurium. The goal is predicting whether each compound possesses mutagenicity effect on the Gram-negative bacterium Salmonella typhimurium.

NCI1 (the anti-cancer activity prediction dataset) is a balanced subset of datasets of chemical compounds screened for activity against non-small cell lung cancer and ovarian cancer cell lines (Wale et al., 2008). It has 37 discrete labels.

PROTEINS (Borgwardt et al., 2005; Dobson & Doig, 2003) dataset consists of proteins represented as graphs. Nodes represent the amino acids and an edge occurs between two vertices if they are neighbors in the amino-acid sequence or in 3D space. It has 3 discrete labels, representing helix, sheet or turn. The aim is to distinguish proteins into enzymes and non-enzymes.

PTC (Predictive Toxicology Challenge) dataset (Helma et al., 2001; Kriege & Mutzel, 2012) is a collection of chemical compounds reporting the carcinogenicity for Male Rats (MR), Female Rats (FR), Male Mice (MM) and Female Mice (FM). We selected graphs of Male Rats (MR) for evaluation, consisting of 344 graphs very small and sparse with 19 discrete labels.

ENZYMES consists of protein tertiary structures obtained from Borgwardt et al. (2005), in particular 600 enzymes from the BRENDA enzyme database (Schomburg et al., 2004). The task for this dataset is assigning each enzyme to one of the 6 EC top-level classes. For our experiments we considered only 3 discrete labels.

IMDB-BINARY dataset (Yanardag & Vishwanathan, 2015) collects graphs about movie collaboration. Each graph consists of nodes representing actors/actress who played roles in movies in IMDB and an edge between two vertices takes place if they appear in the same movie. Graphs are derived from the Action and Romance genres. The task is to identify which genre an ego-network graph belongs to.

