# OpenReview forum: "Graph Kernel Neural Networks"
_ICLR.cc/2022/Conference — ICLR 2022 Submitted_

### Official Review · Reviewer_6pRJ · 2021-10-26

**Correctness:** 4
**Technical Novelty And Significance:** 3
**Empirical Novelty And Significance:** 2
**Recommendation:** 5
**Confidence:** 4

**Main Review:**

- In summary, the approach proposed in this paper is interesting and deals with a topic that has not been thoroughly studied yet. The paper generalizes the model presented in [1] and allows the use of any graph kernel as a component of graph neural networks. However, the paper in its current form does not appear ready for publication and there are several concerns that the authors need to address (see comments below). Furthermore, the empirical results are okay, it seems that the model is competitive with the baselines, but they are not impressive.

- One of the main limitations of the proposed model is its computational complexity, but the authors do not discuss it in the paper. Since the authors extract subgraphs of radius r centered at each node, I would expect the complexity of the model to be high especially when r is set to large values (e.g., 3, 4, etc). I also guess that the high complexity of the model is the reason why the authors do not evaluate it on any of the REDDIT datasets or on COLLAB since these datasets contain very large graphs. I would suggest the authors also report the running time of the proposed model on the different datasets such that one can compare them against those of other baselines. The authors mention in the conclusion that there are no widely available GPU implementations of graph kernels, thus it is likely that the running time of the proposed model would be prohibitive for real-world applications. This is a key weakness of the paper, but still reporting those running times would strengthen the paper.

- It is not clear to me why the input graph is not directly compared against the structural masks as in [1], but instead the authors extract a subraph centered around each node and compare those subgraphs against the masks. I guess that this can lead to higher expressive power (the model is more expressive than 1-WL), however, as discussed above, it also increases the computational complexity of the model.

- In the literature of graph kernels, there exist some kernels that can handle graphs whose nodes are annotated with feature vectors such as the GraphHopper kernel [2] or the multiscale laplacian kernel [3]. I wonder why the authors do not employ those kernels and instead resort to vector quantization techniques which lead to loss of information. Perhaps this is due to the computational complexity of those kernels. I would like the authors to comment on that and it would also be nice if the authors could present some experimental results of the proposed model using one of the above kernels on some dataset that contains small graphs.

- The k-means algorithm is run multiple time at each epoch (once per batch). I am not thus sure whether the clustering result of the k-means algorithm is consistent across different batches. The authors initialize the centroids of the algorithm using the centroids of the previous batch. I would suggest the authors perform an experiment and measure the average distance of each centroid from one batch to the next. If those distances are large, this might indicate that the model is unstable.

- The learning process employed in this paper is different from a traditional learning setting since the model samples an edit operation, and it does not directly learn which operation is the best. I feel that such an approach can easily get stuck in local optima. In such a scenario, the masks could be of no practical interest. I would suggest the authors investigate how much the structure of the masks changes during training. The authors could also borrow ideas from the field of discrete optimization, and design some more sophisticated approach for updating the masks.

- The authors perform their ablation study on the MUTAG dataset. This dataset is very small (188 graph in total) and in general, obtained results are very sensitive to fold assignments and potentially to weight initialization. I am not thus sure whether the results of the ablation study are significant and whether they would also apply to other datasets. The authors should consider to use another dataset that contains a larger number of graphs such that the obtained results are more valid.

- Please specify how the representation of the entire graph is produced. Do the authors use a sum aggregator?

- Typos:
	- p.2: "includes methods where" -> "include methods where"
	- p.3: "are assumes" -> "are assumed"
	- p.6: "G_1 = (V_2, E_2)" -> "G_2 = (V_2, E_2)"

[1] Nikolentzos G. and Vazirgiannis M., "Random Walk Graph Neural Networks". Advances in Neural Information Processing Systems, pp. 16211-16222, 2020.\
[2] Feragen A., Kasenburg N., Petersen J., de Bruijne M. and Borgwardt K.M., "Scalable kernels for graphs with continuous attributes". Advances in Neural Information Processing Systems, pp. 216-224, 2013.\
[3] Kondor R. and Pan H., "The multiscale laplacian graph kernel". Advances in Neural Information Processing Systems, pp. 2990-2998, 2016.

**Summary Of The Paper:**

In this paper, the authors propose a model that uses graph kernels to extend the convolution operator to the graph domain. Given a graph, the model extracts a subgraph centered around each node of the graph and generates a representation for that node by comparing its subgraph against a number of structural masks using graph kernels. Since graph kernels are not in general differentiable functions, the authors use a discrete randomized descent strategy to optimize the structural masks. The authors also add a Jensen-Shannon divergence loss term to force the masks provide dissimilar responses from each other. The proposed model was evaluated on standard graph classification datasets where it was found to be competitive with the baselines.

**Summary Of The Review:**

Overall, this is an interesting paper which studies a topic that has not been fully explored yet.  Even though the originality of the paper is not stellar, the paper introduces some new ideas. However, there are several issues that the authors need to address (e.g., running time, investigate whether clustering is consistent, ablation study on some other dataset, etc). Furthermore, the empirical results are not bad, but they are also not impressive. Thus, the paper does not seem ready for publication in its current state.

---

> ### Author Response · Authors · 2021-11-20
> **Comments on complexity and type of kernels**
>
> *One of the main limitations of the proposed model is its computational complexity, but the authors do not discuss it in the paper. Since the authors extract subgraphs of radius r centered at each node, I would expect the complexity of the model to be high especially when r is set to large values (e.g., 3, 4). I also guess that the high complexity of the model is the reason why the authors do not evaluate it on any of the REDDIT datasets or on COLLAB since these datasets contain very large graphs. I would suggest the authors also report the running time of the proposed model on the different datasets such that one can compare them against those of other baselines. The authors mention in the conclusion that there are no widely available GPU implementations of graph kernels, thus it is likely that the running time of the proposed model would be prohibitive for real-world applications. This is a key weakness of the paper, but still reporting those running times would strengthen the paper.*
>
> Response: As also suggested by the other reviewers, will add a section in the draft (within the rebuttal deadline) discussing the complexity and runtime of our model. We should stress however that, at least as far as graph classification is concerned (and particularly in the case of [bio]-chemical graphs - an extremely relevant real world application), the size of the graphs is fairly limited and thus our model copes well with them in terms of runtime. Large graphs, for example large social networks, are commonly encountered within node classification problems.
>
> REDDIT and COLLAB are indeed large datasets with large graphs that would have taken a seizable runtime to complete training. The relation with the subgraphs radius r is complex and costly and depends on the choice of the kernel. However, the computational complexity of the network grows linearly with the size of the graph itself. Larger graphs result in a proportionally larger number of subgraphs, which for training purposes is equivalent to having a larger number of graphs in the dataset.
>
> The choice of the values of r in the ablation study (2,3,4,5) was simply due to the observation of a clear trend within this range, however our model has no particular issue coping with larger values of r.
>
> ===
>
> *It is not clear to me why the input graph is not directly compared against the structural masks as in [1], but instead the authors extract a subgraph centered around each node and compare those subgraphs against the masks. I guess that this can lead to higher expressive power (the model is more expressive than 1-WL), however, as discussed above, it also increases the computational complexity of the model.*
>
> Response: This scenario would be equivalent to setting a radius r equal to the radius of the input graph itself. This would have the effect of breaking locality which in turn would have the effect of reducing the expressive power of our model, i.e., all nodes will produce the same response. Moreover, considering that the complexity of our model is linear with respect to the number of input graph nodes and polynomial with respect to the subgraph radius r (depending on the choice of kernel), it’s clearly less costly to have a large number of subgraphs with small r, rather than a single large subgraph (equal to the input graph).
>
> Note also that in [1] the authors fix the starting of the random walk on the location of the convolution and, despite not having a fixed radius, a decay factor effectively limits the influence of structures located farther from the convolution center. On the other hand, we create the subgraph around the convolution center but the location information of the convolution center is ignored when computing the kernel between the corresponding subgraph and the input graph.
>
> ===
>
> *In the literature of graph kernels, there exist some kernels that can handle graphs whose nodes are annotated with feature vectors such as the GraphHopper kernel [2] or the multiscale laplacian kernel [3]. I wonder why the authors do not employ those kernels and instead resort to vector quantization techniques which lead to loss of information. Perhaps this is due to the computational complexity of those kernels. I would like the authors to comment on that and it would also be nice if the authors could present some experimental results of the proposed model using one of the above kernels on some dataset that contains small graphs.*
>
> Response: This is a good suggestion that we aim to investigate in future work. In the current experiment we made use of graph kernels implemented in the GraKel python library (https://github.com/ysig/GraKeL). All the graph datasets considered in this paper have discrete labels, and we therefore focused our attention on the WL kernel because this is widely considered a strong baseline. Note that to be able to fully backpropagate and get rid of the vector quantisation we would need the kernel to be differentiable with respect to the node features.

---

> > ### Comment · Reviewer_6pRJ · 2021-11-22
> > **response**
> >
> > I would like to thank the authors for their response and for addressing most of my concerns. However, I am still not convinced about the significance of the paper. The experimental results are promising but not that strong, and considering the high complexity of the method, I wonder what is the main advantage of the proposed model and why someone would be interested in applying it to some real-world application. Based on the above, I will keep my current score.

---

> > > ### Author Response · Authors · 2021-11-23
> > > **On the significance of our paper**
> > >
> > > We respect the reviewer opinion but we do think that our works gives an important contribution by introducing a fully structural model that opens the door for interpretability of the learned masks. We believe this is not just important but in fact essential in real-world applications, and it's therefore bound to attract significant interest. The interpretability and explainability of decisions based on AI models are increasingly seen as being of pivotal importance, especially when personal data is concerned. This is for example regulated by the General Data Protection Regulation (GDPR) in the EU and the Data Protection Act 2018 (DPA 2018) in the UK, which requires one "to be able to give an individual an explanation of a fully automated decision to enable their rights to obtain meaningful information, express their point of view and contest the decision" [1].
> > >
> > > To further support the relevance of the interpretability aspect, we have now added an experimental analysis of the convergence and stability of the learned structural masks in Appendix A.3 (see figures 5 and 6 in the appendix, in particular). This is in addition to the interpretability analysis of section 4.2.
> > >
> > > We do agree that the ability to peek into the black box comes at the cost of increased computational complexity, however we do not believe that this cost renders the application of our model impractical (we have added a discussion of the computational complexity and performed a runtime analysis in section A.4 of the Appendix), nor do we think that this negates the merit of introducing it to the community, as the computational issues can indeed be addressed in the future by using faster kernels, possibly implemented to run on GPUs.
> > >
> > > [1] https://ico.org.uk/for-organisations/guide-to-data-protection/key-dp-themes/explaining-decisions-made-with-artificial-intelligence/part-1-the-basics-of-explaining-ai/legal-framework/

---

> ### Author Response · Authors · 2021-11-20
> **Comments on centroids, learning process, and other details**
>
> *The k-means algorithm is run multiple time at each epoch (once per batch). I am not thus sure whether the clustering result of the k-means algorithm is consistent across different batches. The authors initialize the centroids of the algorithm using the centroids of the previous batch. I would suggest the authors perform an experiment and measure the average distance of each centroid from one batch to the next. If those distances are large, this might indicate that the model is unstable.*
>
> Response: We thank the reviewer for this interesting suggestion. We verified the displacement of the centroids induced by the k-means clustering in different models and datasets. With a batch size of 16, at convergence, the displacement is always within 0.9% of the average distance between the centroids.
>
> ===
>
> *The learning process employed in this paper is different from a traditional learning setting since the model samples an edit operation, and it does not directly learn which operation is the best. I feel that such an approach can easily get stuck in local optima. In such a scenario, the masks could be of no practical interest. I would suggest the authors investigate how much the structure of the masks changes during training. The authors could also borrow ideas from the field of discrete optimization, and design some more sophisticated approach for updating the masks.*
>
> Response: Local minima are indeed a widespread problem in all deep learning models, however empirical evidence shows that this is not necessarily an issue. As can be seen from the qualitative experiments, the learned masks do have a rather good interpretability, which in turn suggests stability of the learning process. We will include a discussion and experiment to show the stability of the learned masks under different model initialisations.
>
> ===
>
> *Please specify how the representation of the entire graph is produced. Do the authors use a sum aggregator?*
>
> Response: Indeed, we use a sum aggregator. We will make this clearer in the text of the revised draft.
>
> ===
>
> *Typos*
>
> Response: all fixed.

---

### Official Review · Reviewer_55BS · 2021-11-02

**Correctness:** 3
**Technical Novelty And Significance:** 4
**Empirical Novelty And Significance:** 3
**Recommendation:** 6
**Confidence:** 4

**Main Review:**

Strengths:
- The paper presents a novel and interesting class of learning algorithms for processing graph-based data. This is is a particularly relevant area in machine learning which contributes to raising the significance of this work.
- The model is somewhat more interpretable than competitors and seems to be well apt to settings where the topological structure of the graphs plays a major role
- The paper is generally well written and easy to follow, although some parts of the presentation could be improved as in the detailed comment 1.

Weaknesses:
- The model has some limitations. First, it seems limited to graph classification/regression problems. Could it be used also for node-level tasks? Furthermore, it is suitable to settings in which nodes have either no or only a discrete set of labels (with low cardinality). The authors should be more clear about these limitations from the introduction.
- The optimization problem involves optimizing over discrete structures. I believe the part regarding the learning technique could be clearer, more formal and with more details. Please see detailed comments. For this reason, I have some doubts about how much of the performance is driven by learning and how much by engineering.
- The experimental section is improbable, see detailed comments 4.

Detailed comments:
1. Clarity:
a) in *To keep the labelling consistent among different batches, the k-means algorithm centroids at step $\ell + 1$ are initialized using the centroids
computed at the previous step* (pag 4) does batch mean layer? Or is it about batched (stochastic?) processing of the nodes or of graphs in a mini-batch of graphs?
b) On page 3, it is not entirely clear to me what $g$ is and what $\theta$ and $\Theta$ refer to (specifically, do they represent the same sets of parameters?). Does (1) implements $g_{\theta}$?I think this passage could be written much more clearly.
b2) This confusion is carried forward also to eq (4). What is $\theta$ there precisely? Only the parameters of the last classification layer?
c)  The $h$ in (4) seems to be quite different from the $h$ in (1), probably it would be better to change notation there to avoid confusion.
d) The proofs are only sketched, with little details and not particularly easy to follow. Just to give an example, in *the presence of a triangle in the graph induces the existence of at least three sub-graphs of radius 1 with *cross-neighbor* connections.* what does cross-neighbor specifically mean?

2. I never heard of the discrete randomized descent method and the authors seem not to make any reference. Can the authors comment on this? If this is a (well-known) approach, please add a reference, otherwise, I think there should be more explanation about the method. For instance, in P(6) there is an equality sign which seems misplaced (also the authors talk about "estimation"). On the same line, I think also the concept of discrete sub-gradient is non-standard and needs to be defined or linked to some relevant literature. For instance, it seems that it differs from the concept in [1].

3. The authors argue that in order to backpropagate the error in a multi-layer version of the architecture they need to add skip connections since they do not propagate the gradient through the vector quantization operation. However "discrete gradients" are used to learn both the mask structures and the edit distributions. Can't they (or similar strategies) be used also to propagate the gradients throw the quantization operations?

 4. Regarding experimental validation: a) I would like to see a training curve. How much does the model improve with iterative optimization? How many iterations are needed and how many passes through the training data? b) I wonder what's the effect of the initialization. I would ideally like to see a set of experiments to ascertain how much the (random) initialization impacts the performances and convergence/training time. c) I believe that grid search makes it really difficult to understand what is it the tuning effort of the proposed model (learning algorithm)  with respect to the other competitors. This is because the total allocated computational budget can vary extensively. In my opinion, random search would have been a better and equally simple alternative, as it allows to have an "anytime solution" which makes it easier to compare methods (i.e. at each timestep, one could compare the best model found so far.

Minor points:

5. The authors make the point that GKN can leverage the presence of many graph kernels that have been proposed in the literature. However, most of the experiments have been done with the WL kernel. Would it be possible to have a GKN that uses multiple graph kernels? If so, how would it fare on the benchmark datasets?

6. A discussion of the computational complexity is entirely missing, as well as reports of runtime measurements.

7. I would call section 4.1*Sensitivity study* rather than *Ablation study*. Ablation usually refers to the practice of removing components of the learning algorithm that is being proposed. Instead, section 4.1 is about seeing what is the relative effect of each of the model hyperparameters.

8. Missing sensitivity analysis of the number of clusters to use during the quantization step.

[1] Bagirov, Adil M., Bülent Karasözen, and Meral Sezer. "Discrete gradient method: derivative-free method for nonsmooth optimization." Journal of Optimization Theory and Applications 137.2 (2008): 317-334.


**Summary Of The Paper:**

This paper proposes a novel architecture for graph processing that uses graph kernels with learnable structural masks.
The core feature of this novel architecture is the graph kernel convolution (GKC) layer.
The GKC layer consists of computing for each node $v$ a subgraph centred in $v$ of radius $r$, then the subgraph is compared with a series of structural masks through a graph kernel. The resulting kernel responses are collected into a vector, which is then quantized using k-means clustering, and the cluster label is used as the new node label.
Multiple layers can be stacked, although they require adding skip connections because of the quantization operation.
The authors propose also

**Summary Of The Review:**

This paper is a potentially significant contribution to the area of graph ML.
At the moment, however, I have some concerns regarding the training mechanism proposed, specifically regarding the utilization of "discrete gradients". Moreover, the experimental section could be improved.

---

> ### Author Response · Authors · 2021-11-20
> **Comments on node-level tasks and optimisation strategy**
>
> *The model has some limitations. First, it seems limited to graph classification/regression problems. Could it be used also for node-level tasks? Furthermore, it is suitable to settings in which nodes have either no or only a discrete set of labels (with low cardinality). The authors should be more clear about these limitations from the introduction.*
>
> Response:
> In principle, our model is general and thus applicable to node-level tasks as well. However, the fundamental goal of our model is to reconstruct the structural information needed for classification and is thus particularly suited for problems where the structure plays a pivotal role. Indeed, in most graph classification problems the structure provides the most relevant information for classification, with the node/edge features taking a secondary role. The situation is usually reversed for node classification problems (often concerning social interactions) , where the distribution of (high-dimensional) continuous/mixed features across the immediate neighbours of a node hold most of the information needed to classify the node itself.
>
> Furthermore, with our choice of graph kernels, which was driven by the structural nature of the problem at hand, our model inevitably has a hard time trying to handle high-dimensional features further placing the approach at a disadvantage for node classification.
>
> It would indeed be possible to adapt our model for node-level tasks by using kernels that can handle continuous attributes and that are not necessarily fully differentiable, but at least differentiable with respect to the node features, thus removing the need of the vector quantisation step.
>
> We will edit the introduction of the draft within the rebuttal deadline to stress the limitations of the current version of our architecture.
>
> ===
>
> *Comments on clarity*
>
> Response:
> a) Batch means mini-batch, we will make it clearer in the revised draft.
>
> b, b2, c) Eq. 3 defines the general formulation of the message passing convolution. We will fix the notation and make it clearer in the revised draft.
>
> d) We will revise and clarify the text of the sketches of proof to make sure all the necessary terms are properly defined.
>
> ===
>
> *I never heard of the discrete randomized descent method and the authors seem not to make any reference. Can the authors comment on this? If this is a (well-known) approach, please add a reference, otherwise, I think there should be more explanation about the method. For instance, in P(6) there is an equality sign which seems misplaced (also the authors talk about "estimation"). On the same line, I think also the concept of discrete sub-gradient is non-standard and needs to be defined or linked to some relevant literature. For instance, it seems that it differs from the concept in [1].*
>
> Response: We agree that the name is indeed not a standard one. We believe that we discussed the method at length in Section 3.4 and Appendix A.1, however we would like to understand better what the reviewer finds confusing: for example, what is the “P(6)” referenced in the comment? What we mean by “estimation” is explained in Appendix A.1. By “sub-gradient” we mean the gradient computed over the subset of the coordinates randomly selected by the algorithm. We will clarify this in the text.
>
> ===
>
> *The authors argue that in order to backpropagate the error in a multi-layer version of the architecture they need to add skip connections since they do not propagate the gradient through the vector quantization operation. However "discrete gradients" are used to learn both the mask structures and the edit distributions. Can't they (or similar strategies) be used also to propagate the gradients throw the quantization operations?*
>
> Response: Currently in the computation of the gradient for the node features, we only compute the gradient of the kernel value with respect to the masks of the current layer, and not with respect to the input graph for that layer, which would be required to backpropagate the gradient to the previous layer. This is due to the presence of the vector quantization layer which complicates the backpropagation with respect to node features. As a consequence node features are updated only along the skip connection. Empirically we observed this not to be a major problem as longer paths have diminishing weight.
> This makes the learning process much faster as we do not need to compute every kernel for each edit operation, but only the ones directly affected by the modified mask.
> We will make this clear in the revised text.

---

> > ### Comment · Reviewer_55BS · 2021-11-24
> > **Thank you**
> >
> > I thank the author for their rebuttal, which addresses some of my concerns. I will edit my review and possibly score after a discussion stage.

---

> ### Author Response · Authors · 2021-11-20
> **Comments on experimental validation and minor points**
>
> *Regarding experimental validation: [...]*
>
> Response:
> a) We thank the reviewer for suggesting this additional experiment. We will include in the Appendix of the draft.
> b)  In our experiments we didn’t observe a significant impact of the random initialization of the structural masks on the network performance nor on the final learned masks. We will add to the draft a qualitative visualization showing the similarity of the learned masks between different training runs of the same model. We are happy to include more results about this in the camera ready, but unfortunately the expensive nature of the computations doesn’t allow us to perform them in time for the end of this rebuttal phase.
> c) We thank the reviewer for the suggestion. We simulated Grid Search on all the datasets for a subset of the methods (GraphSAGE, sGIN, RWNN, and Ours). Our method has a similar behavior to that of our competitors, showing a benefit when including more models in the evaluation. The average drop in performance from randomly selecting 5 models in contrast to performing a full grid search is around 2% and decreases with the number of models considered. We will add the results to the draft.
>
> ===
>
> *The authors make the point that GKN can leverage the presence of many graph kernels that have been proposed in the literature. However, most of the experiments have been done with the WL kernel. Would it be possible to have a GKN that uses multiple graph kernels? If so, how would it fare on the benchmark datasets?*
>
> Response: It would indeed be possible to have a GKN that uses multiple graph kernels. Following this comment, we have tried to use multiple kernels with the current implementation of the model, however the performance was slightly below that of the model with the best kernel, likely due to the fact that the learned masks were shared across all the kernels, thus limiting the ability to capture the specific structural features each kernel is sensitive to.
>
> ===
>
> *A discussion of the computational complexity is entirely missing, as well as reports of runtime measurements.*
>
> Response: We will add a section about this in the draft.
>
> ===
>
> *I would call section 4.1Sensitivity study rather than Ablation study. Ablation usually refers to the practice of removing components of the learning algorithm that is being proposed. Instead, section 4.1 is about seeing what is the relative effect of each of the model hyperparameters.*
>
> Response: We will edit the section title accordingly.
>
> ===
>
> *Missing sensitivity analysis of the number of clusters to use during the quantization step.*
>
> Response: In this paper we made the decision of setting the number of clusters equal to the number of structural masks. This is a reasonable choice since the kernels measure the similarity to the masks and we expect the masks to be dissimilar to each other. As a consequence, we expect the resulting output feature vector to be similar to a one-hot encoding of the most similar mask. We will edit the draft to clarify this.

---

### Official Review · Reviewer_ovv9 · 2021-11-03

**Correctness:** 3
**Technical Novelty And Significance:** 3
**Empirical Novelty And Significance:** 2
**Recommendation:** 6
**Confidence:** 3

**Main Review:**

The paper is well motivated and easy to follow. The authors provide a detailed description of the problem and related works and background.


Regarding contribution #1 "Unlike existing approaches that require embedding the input graph
into a larger, relaxed space with a higher likelihood of ending up in a local minimum, our model
is fully structural". -- Why is this true? Usually, we would like to have a high dimensional feature space, to capture more abstract features of the input. I am not convinced that remaining in a smaller dimension is more beneficial. Perhaps the authors can elaborate on this point ?

Regarding the second paragraph in the related works section, where diffusion in GNNs is discussed - while the statement is correct, it was also recently shown in "PDE-GCN: Novel Architectures for Graph Neural Networks Motivated by Partial Differential Equations" (https://arxiv.org/abs/2108.01938) that not all kind of applications and datasets indeed benefit from diffusion, and at times propagation is favored. I believe that discussion that in the paper will contribute to its content.

All the experiments consider graph classification, which is indeed an important task. However, it would be interesting to know how the proposed method would work, for instance, for node classification (e.g., Cora, Citeseer). Also, it would be interesting to know if the method can be used for geometric datasets, where a more learned masks may be easier to explain.

In all experiments, a rather shallow network is considered to my understanding. Either 1,2,3 kernel layers are used, following a classifier. Since oversmoothing is a known phenomenon in typical GCNs, it would be interesting to know if the proposed method can prevent this problem. It may be, that learning the correct masks can aid in feature preservation and distinguishability (this is also related to the diffusion/propagation discussion above and the works cited by the authors).

Overall, the results presented in Table 1 show that in some cases the proposed method outperforms other by quite a small margin, but in most cases it is not better methods.

The authors state that graph kernels are limited by implementation. While this may be fine, I think that the authors can strengthen their claims by adding some information regarding the run times of their work compared to others.







**Summary Of The Paper:**

The authors propose a method that is based on graph kernels to perform graph convolution, which add to the interpretability of GNNs.
A framework for learning graph kernels is introduced and studied on multiple datasets. The results show improvement in some datasets.

**Summary Of The Review:**

The paper is clearly written and motivated but lacks on the experimental side, both in results of current experiments, and also the scope of the experiments is rather limited.

---

> ### Author Response · Authors · 2021-11-20
> **Comments regarding unconvincing claim on issues with higher dimensional space + citation to add + node classification**
>
> *Regarding contribution #1 "Unlike existing approaches that require embedding the input graph into a larger, relaxed space with a higher likelihood of ending up in a local minimum, our model is fully structural". -- Why is this true? Usually, we would like to have a high dimensional feature space, to capture more abstract features of the input. I am not convinced that remaining in a smaller dimension is more beneficial. Perhaps the authors can elaborate on this point?*
>
> Response: Indeed, a continuous high dimensional embedding space has the effect of increasing the complexity of the hypothesis space, thus allowing one to seek more complex and arbitrary decision boundaries. While this can allow us to discriminate more complex problems, it also has the potential of overfitting the problem and creating more local optima. For example, it has been observed in the literature that GCN are prone to overfit and need the use of dropout strategies to reach good performance [1,2,3]. Indeed, we have also seen in our preliminary analysis a significant drop in classification accuracy for the GCN model on the MUTAG dataset if dropout is omitted.
>
> One may argue that given sufficient data, time, and a suitable optimisation strategy, the issue of the local optima can be minimised. However this is not always possible, often due to lack of data (an issue often found in graph-based machine learning, where data augmentation is not as trivial as in the image domain).
>
> The experimental results do agree with our intuition, showing a clear improvement wrt GCNs and thus suggesting that restricting the search space to discrete structures has an advantage in terms of avoiding local optima. In addition to this, keeping the model fully structural has the clear benefit of allowing the interpretation of the convolutional filters in terms of sub-structures of interest.
>
> We will revise this sentence in the draft accordingly, which will be updated within the 22nd of November.
>
> [1] Cong, Weilin, et al.. "On Provable Benefits of Depth in Training Graph Convolutional Networks." NeurIPS (2021).
> [2] Rong, Yu, et al. "Dropedge: Towards deep graph convolutional networks on node classification." arXiv preprint arXiv:1907.10903 (2019).
> [3] Zhou, Kuangqi, et al. "Effective training strategies for deep graph neural networks." arXiv e-prints (2020): arXiv-2006.
>
> ===
>
> *Regarding the second paragraph in the related works section, where diffusion in GNNs is discussed - while the statement is correct, it was also recently shown in "PDE-GCN: Novel Architectures for Graph Neural Networks Motivated by Partial Differential Equations" (https://arxiv.org/abs/2108.01938) that not all kind of applications and datasets indeed benefit from diffusion, and at times propagation is favored. I believe that discussion that in the paper will contribute to its content.*
>
> Response: We thank the reviewer for pointing out the relevance of this paper, we will discuss it in the related work section of the revised draft.
>
> ===
>
> *All the experiments consider graph classification, which is indeed an important task. However, it would be interesting to know how the proposed method would work, for instance, for node classification (e.g., Cora, Citeseer). Also, it would be interesting to know if the method can be used for geometric datasets, where a more learned masks may be easier to explain.*
>
> Response: Although our model is general in principle (and thus applicable to node classification), the fundamental goal of our model is to reconstruct the structural information needed for classification and is thus particularly suited for problems where the structure plays a pivotal role. Indeed, in most graph classification problems the structure provides the most relevant information for classification, with the node/edge features taking a secondary role. The situation is usually reversed for node classification problems (often concerning social interactions), where the distribution of (high-dimensional) continuous/mixed features across the immediate neighbours of a node hold most of the information needed to classify the node itself.
>
> Furthermore, with our choice of graph kernels, which was driven by the structural nature of the problem at hand, our model inevitably has a hard time trying to handle high-dimensional features further placing the approach at a disadvantage for node classification.
>
> Moreover, in geometric datasets the structure (which is induced by the geometric information) is of secondary importance and it varies depending on the manifold sampling strategy. Once again, our model seeks to identify the presence of discriminative structures and is therefore not ideally suited for this scenario.
>
> We will edit the introduction of the draft accordingly to clarify these points.

---

> ### Author Response · Authors · 2021-11-20
> **Other comments**
>
> *In all experiments, a rather shallow network is considered to my understanding. Either 1,2,3 kernel layers are used, following a classifier. Since oversmoothing is a known phenomenon in typical GCNs, it would be interesting to know if the proposed method can prevent this problem. It may be, that learning the correct masks can aid in feature preservation and distinguishability (this is also related to the diffusion/propagation discussion above and the works cited by the authors).*
>
> Response: Indeed, in our ablation study we did actually investigate the variation of the accuracy wrt the number of layers for depth greater than 3. We don’t see any clear evidence of over-smoothing as the accuracy doesn’t dramatically drop when increasing the layers as observed in GCNs. Due to the limited time available for the rebuttal we are not able to complete the full experiment, but we will include an analysis of deeper models in the camera ready version of the paper.
>
> ===
>
> *Overall, the results presented in Table 1 show that in some cases the proposed method outperforms other by quite a small margin, but in most cases it is not better methods.*
>
> Response: It should be noted that our architecture is relatively simple, the closest alternative being the standard GCN. With respect to this, we perform significantly better. With respect to other more complex yet better performing methods, our model has the advantage of having interpretable filters. We believe this is an important feature in a world that is increasingly trying to move away from high performing black-box solutions, in favour of well performing yet more interpretable models.
>
> ===
>
> *The authors state that graph kernels are limited by implementation. While this may be fine, I think that the authors can strengthen their claims by adding some information regarding the run times of their work compared to others.*
>
> Response: We will add to the draft a new subsection that discusses the complexity and runtime of our architecture. In particular, we will show the average training time per dataset (both total and per epoch) of the models trained during the grid search and the average inference time of the best model.

---

### Author Response · Authors · 2021-11-23
**To All Reviewers: Updated Manuscript and Appendix**

We thank the reviewers for their comments and suggestions that helped us to improve the manuscript. We submitted a revised version highlighting the modified parts in red. Some of the main changes include:
1) we revised the motivation and the scope of this work in the introduction part;
2) we clarified the multilayer architecture and verified the stability of the k-means clustering;
3) we added an experimental analysis of the convergence and stability of the learned structural masks in Appendix A.3;
4) we discussed the computational complexity and performed a runtime analysis in Appendix A.4;
5) we analyzed the impact of a random grid search hyperparameters optimization on the classification accuracy in Appendix A.5;
6) we fixed the mathematical notation and corrected other minor spelling/grammatical mistakes.

All revisions have been highlighted in red for the reviewers' convenience.

---

### Decision · Program_Chairs · 2022-01-20

**Decision:**

Reject

**Comment:**

The paper uses graph kernels to perform local convolutions and achieve better expressiveness than classical GNNs. The paper received three borderline reviews. The area chair found the feedback to be consistent and constructive and agrees with most statements made by the reviewers. Overall, the idea has some interest (even though there are other works who also propose hybrid approaches between graph kernels and GNNs, as noted in the paper). Nevertheless, there is a lot of room for improvement regarding the experimental validation and the results are not very convincing (yet?). The datasets used in the paper have been traditionally used for evaluating GNNs but they have strong limitations due to their small size and it is often hard to draw conclusions from them. If the method does not suffer from scalablity issues, it is likely that more interesting results could be obtained by using ZINC or MOLHIV datasets, which are larger and often provide statistically significant results.

Overall, these issues may require a major revision and unfortunately, the area chair believes that the paper is not ready for publication.